# Image Embeddings Extracted from CNNs Outperform Other Transfer Learning Approaches in Classification of Chest Radiographs

**DOI:** 10.3390/diagnostics12092084

**Published:** 2022-08-28

**Authors:** Noemi Gozzi, Edoardo Giacomello, Martina Sollini, Margarita Kirienko, Angela Ammirabile, Pierluca Lanzi, Daniele Loiacono, Arturo Chiti

**Affiliations:** 1IRCCS Humanitas Research Hospital, Via Manzoni 56, Rozzano, 20089 Milan, Italy; 2Laboratory for Neuroengineering, Department of Health Sciences and Technology, Institute for Robotics and Intelligent Systems, ETH Zurich, 8092 Zurich, Switzerland; 3Dipartimento di Elettronica, Informazione e Bioingegneria, Via Giuseppe Ponzio 34, 20133 Milan, Italy; 4Department of Biomedical Sciences, Humanitas University, Via Rita Levi Montalcini 4, Pieve Emanuele, 20090 Milan, Italy; 5Fondazione IRCCS Istituto Nazionale Tumori, Via G. Venezian 1, 20133 Milan, Italy

**Keywords:** medical imaging, X-rays, artificial intelligence, transfer learning, explainability

## Abstract

To identify the best transfer learning approach for the identification of the most frequent abnormalities on chest radiographs (CXRs), we used embeddings extracted from pretrained convolutional neural networks (CNNs). An explainable AI (XAI) model was applied to interpret black-box model predictions and assess its performance. Seven CNNs were trained on CheXpert. Three transfer learning approaches were thereafter applied to a local dataset. The classification results were ensembled using simple and entropy-weighted averaging. We applied Grad-CAM (an XAI model) to produce a saliency map. Grad-CAM maps were compared to manually extracted regions of interest, and the training time was recorded. The best transfer learning model was that which used image embeddings and random forest with simple averaging, with an average AUC of 0.856. Grad-CAM maps showed that the models focused on specific features of each CXR. CNNs pretrained on a large public dataset of medical images can be exploited as feature extractors for tasks of interest. The extracted image embeddings contain relevant information that can be used to train an additional classifier with satisfactory performance on an independent dataset, demonstrating it to be the optimal transfer learning strategy and overcoming the need for large private datasets, extensive computational resources, and long training times.

## 1. Introduction

The world’s population increased by about threefold between 1950 and 2015 (from 2.5 to 7.3 billion), and this trend is projected to continue in the coming decades (a population of 19.3 billion people is expected in 2100), with a growing share of the aging population (≥65 years) (https://www.eea.europa.eu/data-and-maps/indicators/total-population-outlook-from-unstat-3/assessment-1 (accessed on 30 April 2021)). This projected trend is strongly linked to the increasing demand for medical doctors, including imagers. The medical community has offered some warnings about the urgent need to act (https://www.rcr.ac.uk/press-and-policy/policy-priorities/workforce/radiology-workforce-census (accessed on 30 April 2021)), suggesting that artificial intelligence (AI) might partially fill this gap [1]. The joint venture between AI and diagnostic imaging relies on the advantages offered by machine learning approaches to the medical field, which include the automation of repetitive tasks, the prioritization of unhealthy cases requiring urgent referral, and the development of computer-aided systems for lesion detection and diagnosis [2]. Nonetheless, the majority of such AI-based methods are still research prototypes, and only a few have been introduced in clinical practice [3], despite increasing evidence the superior performance of AI relative to that of doctors [4,5]. A number of reasons may be called upon to explain this fact [6,7,8]. A successful AI-based tool relies on three main ingredients: an effective algorithm, high computational power, and a reliable dataset. Whereas the first two ingredients are generally available and can leverage several applications in different domains, the latter is perhaps the most critical in medical imaging. An adequate quality and amount of data necessary for machine learning approaches are still challenging or unfeasible in most clinical trials [6]. Accordingly, some strategies can be used to cross the hurdle of datasets in the medical imaging field. These comprise virtual clinical trials [9,10], privacy-preserving multicenter collaboration [11], and transfer learning approaches [12]. In particular, transfer learning, i.e., leveraging patterns learned on a large dataset to improve generalization for another task, is an effective approach for computer vision tasks on small datasets. Besides enabling training with a smaller amount of data, avoiding overfitting, transfer learning has shown remarkable performance in generalizing from one task and/or domain to another [13]. However, the optimal transfer learning strategy has not yet been defined due to the lack of dedicated comparative studies. In this work, we propose:Identification of the best transfer learning approach for medical imaging classification, encompassing three steps: (1) pretraining of CNN models on a large publicly available dataset, (2) development of multiple transfer learning methods, and (3) performance evaluation and comparison;Interpretation of CNN black-box predictions using explainable AI (XAI) on a population level and randomly selected set of examples.

We tested this proof-of-concept approach on chest radiographs (CXRs). CXR is the most frequently performed radiological examination. Thus, the semiautomatic interpretation of CXRs could significantly impact medical practice by potentially offering a solution to the shortage of radiologists.

## 2. Materials and Methods

### 2.1. Datasets

The experimental analysis discussed in this paper involved two datasets: (i) CheXpert, a large public dataset, which was used to pretrain several classification models; and (ii) HUM-CXR, a smaller local dataset, which was used to evaluate the investigated transfer learning approaches.

CheXpert. This dataset comprises 224316 CXRs of 65,240 patients collected from the Stanford Hospital from October 2002 to July 2017 [14]. For this study, 191027 CXRs from the original dataset that presented a full reported diagnosis were selected. Each image was annotated with a vector of 14 labels corresponding to major findings in a CXR. Mentions of diseases were extracted from radiology reports with an automatic rule-based system and mapped—for each disease—with positive, negative, and uncertain labels according to their level of confidence. Table 1 shows the data distribution among the 14 labels included in the dataset.

HUM-CXR. We retrospectively collected all chest X-rays performed between 1 May 2019 and 20 June 2019 from the IRCCS Humanitas Research Hospital institutional database. We excluded records (1) not focused on the chest, (2) without images stored in the Institutional PACS, (3) without an available medical report, and (4) without an anteroposterior view. HUM-CXR is composed of 1002 CXRs, including anteroposterior, lateral, and portable (i.e., in bed) CXRs. Labels were manually extracted from medical reports (CJ). Uncertain cases were reassessed by two independent reviewers (M.S. and M.K.), and discordant findings were solved by consensus. Each image was annotated as normal or abnormal; abnormalities were further specified as mediastinum, pleura, diaphragm, device, other, gastrointestinal (GI), pneumothorax (PNX), cardiac, lung, bone, or vascular, resulting in a vector of 12 labels. It was not possible to use available automatic labelers [14] because they are designed for English-language use, whereas our radiological reports were written in Italian. Mediastinum, diaphragm, other, GI, and vascular labels were not included in this work due to a limited number of available X-rays (<30) and significant inconsistencies with CheXpert labels. Ultimately, 941 CXRs were included in the analysis. Table 2 shows the data distribution of the labels selected for this study.

This study was approved by the Ethical Committee of IRCCS Humanitas Research Hospital (approval number 3/18, amendment 37/19); due to the retrospective design, specific informed consent was waived.

Preprocessing. For both datasets, we selected only anteroposterior images. Concerning CheXpert, following the approach described in [15], we resized the images to 256 × 256, and a chest region of 224 × 224 was extracted using a template-matching algorithm. We then normalized the images by scaling their values in the range [0, 1]; because the original models were pretrained on ImageNet, we further standardized them with respect to ImageNet mean and standard deviation. Concerning HUM-CXR, we selected X-rays acquired with an anteroposterior view, screening the images according to the series description in DICOM format, which had to be anteroposterior, posteroanterior, or portable; the final sample comprised 941 image of 746 patients. First, we clipped pixel values with a maximum threshold of 0.9995 quantile to minimize the noise due to the landmark (see Figure 1).

### 2.2. Pretraining on CheXpert

In this work, we trained several classifiers on the CheXpert dataset to predict CXR findings. Following the protocol described in [15], we considered seven convolutional neural networks (CNNs) with different topologies and numbers of parameters: DenseNet121 (7M parameters), DenseNet169 (12,5M parameters), DenseNet201 (18M parameters) [16], InceptionResNetV2 (54M parameters) [17], Xception (21M parameters) [18], VGG16 (15M parameters) [19], and VGG19 (20M parameters) [19]. We selected these seven network architectures because (i) they are the most common architectures used to perform classification, and (ii) the performance of each architecture differed depending on the labels. With no predominant architectures, aggregating multiple models can improve the final performances. To use these networks as classifiers, we removed the original dense layer and replaced it with a global average pooling (GAP) [20] layer, followed by a fully-connected layer with a number of outputs that matched the number of labels. These seven networks were not trained from scratch; instead, following a common practice in CNN training, we performed a first transfer learning step by initializing the convolutional layers of the networks with weights of pretrained models on the ImageNet dataset [21]. Then, we trained all the weights (both convolutional and classification layers) on the CheXpert dataset, using 90% of the sample for training and 10% for validation (further details on the training process can be found in our previous work [15]).

Once trained to classify images, the convolutional blocks of CNN models can be employed as a mean to extract a vector of features from images, usually called image embedding. CNNs learn to classify images by learning an effective input representation directly from raw data; the sequence of convolutional layers progressively reduces the size of the input and extracts features from images from low-level features (e.g., edges, pixel intensities, etc.) in early convolutional layers to high-level semantic features in the latest convolutional layers. Accordingly, the last convolutional block, resulting from the training process, is designed to output a vector with the relevant features.

### 2.3. Transfer Learning

In this paper, we propose three transfer learning approaches, as depicted in Figure 2.

As a reference standard, we mapped the CheXpert labels to the HUM-CXR labels and using the pretrained CNNs. The first transfer learning approach consisted of combining the outputs of pretrained CNNs using stacking [22]. The second approach exploited the pretrained CNNs to compute the image embeddings from HUM-CXR data and used them to train tree-based classifiers. The last approach consisted of tuning the CNNs pretrained on CheXpert on HUM-CXR data. In the remainder of this section, we describe these four approaches in detail.

Pretrained CNNs. This was the most straightforward of the investigated approaches and was used mainly as a baseline. It consisted of providing the HUM-CXR images as input to the CNNs trained on CheXpert and using the output of the networks to classify them based on a mapping between the labels of the two datasets. Table 3 shows the mapping designed as a result of an analysis of the images and labels in the two datasets.

For multiple labels, we selected the maximum output probability of the network for CheXpert labels as the predicted value for the respective HUM-CXR outcome. As reported in previous works [15,23], none of the trained CNNs outperformed any of the other networks on the label problem. Thus, to improve the overall classification performances, we combined the outputs of the trained CNSs through two ensemble methods: simple average and entropy-weighted average. In the case of simple average, the predictions of the classifiers were combined as:(1)$$\tilde{y_{i}}=\frac{1}{N}\overset{N}{\underset{k=1}{\sum }}p_{k,i}$$
where *p_k_*_,*i*_ is the prediction of classifier *k* for label *i*, *N* is the number of classifiers, and yei is the resulting prediction of the ensemble for label *i*.

When using entropy-weighted average, the predictions were combined as:(2)$$\tilde{y_{i}}=\overset{N}{\underset{k=1}{\sum }}(1-H(p_{k,i}))p_{k,i}$$
where *p_k_*_,*i*_ is the prediction of classifier *k* for label *i*, *N* is the number of classifiers, *H*(*p*) = −*p*log_2_(p) − (1 − *p*)log_2_(1 − *p*) is the binary entropy function, and $\tilde{y_{i}}$ is the resulting prediction of the ensemble for label *i*.

Stacking. This approach extends the previous approach by using a method called stacked generalization or stacking [22]. Instead of combining the outputs of the CNNs with a simple or an entropy-weighted average as described above, we combined them using a metaclassifier trained for this purpose. Thus, we trained a random forest (RF) to predict the label for HUM-CXR samples based on the predictions of the seven CNNs trained on CheXpert and mapped to labels of HUM-CXR, as shown in Table 3. The data were divided into a training set (70%) and a test set (30%).

Tree-based classifiers. This approach exploits the CNNs trained on CheXpert to compute the image embeddings of CXRs included in the HUM-CXR dataset. Image embeddings can be used to predict the label of the corresponding images using much simpler models than CNNs, such as tree-based models. The benefit of using tree-based models with respect to CNNs is that they do not require either high computational resources or extremely large datasets for training, making them suitable for smaller single-institution datasets. In this work, we focused on three kinds of tree-based methods: decision tree (DT), random forest (RF), and extremely randomized trees (XRT). For each method, we trained seven classifiers using the seven CNNs pretrained on CheXpert to compute the image embeddings from the HUM-CXR dataset, with 70% of the samples used for training and 30% for testing. As previously mentioned, for the pretrained CNNs, we applied ensemble methods, i.e., the simple average and the entropy-weighted average, to combine these seven classifiers. We tuned the training hyperparameters of the tree-based classifiers with a grid-search optimization using stratified K-fold cross validation (Table 4 shows the parameters).

The results were combined with simple average and entropy-weighted average.

Fine-tuning. This is a common transfer learning approach in deep learning that consists of adapting and retraining the last layers of a pretrained neural network on different data or tasks [13]. Therefore, we removed the fully connected classification layer from the seven CNNs trained on CheXpert and replaced it with a seven-output layer that matched the HUM-CXR labels. Then, the HUM-CXR (70% training, 10% validation) dataset was used to finetune the original networks. The models were fine-tuned for five epochs with early stopping on the validation AUC set to three epochs. Binary cross entropy was used as loss function, and the learning rate was initially set to 1 × 10^−4^, to be reduced by a factor of 10 after each epoch. For each CNN, the best-performing model upon validation was tested on the remaining 20% of the HUM-CXR dataset. The performances were evaluated with simple average, entropy-weighted average, and stacking.

### 2.4. Performance Assessment

To assess the performances of our classifiers, we computed the area under the receiving operating characteristic (ROC) curve. The ROC curve was obtained by plotting the true positive rate (TPR) (or sensitivity) versus the false positive rate (FPR) (or 1-specificity). Values higher than 0.8 were considered excellent [24], and the training time was recorded.

### 2.5. Explainability

Despite having proven successful predictive performance, CNNs are recognized as black-box models, i.e., the reasoning behind the algorithm is unknown or known but not interpretable by humans. In order to build trust in AI systems, it is necessary to provide the user with details and reasons to make their functioning clear or easy to understand [25]. We applied gradient-weighted class activation map (Grad-CAM) [26], a state-of-the-art class-discriminative localization technique for CNN interpretation that outputs a visualization of the regions of the input (heat map) that are relevant for a specific prediction. Grad-CAM uses the gradient of an output class in the final convolutional layer to produce a saliency map that highlights areas of the image relevant to detection of the output class. Then, the map is upsampled to the dimensions of the original image, and the mask is superimposed on the CXR. Grad-CAM is considered an outcome explanation method, providing a local explanation for each instance. Therefore, we applied Grad-CAM to randomly selected HUM-CXR data. Grad-CAM heat maps were computed for each CNN model and averaged. In addition to superimposing them on the original image, we used Grad-CAM heat maps to automatically generate a bounding box surrounding the area associated with the outcome. We created a mask with the salient part of the heat map (pixel importance larger than the 0.8 quantile) and used its contours to draw a bounding box highlighting the region of the input that contributed most to the prediction. Grad-CAM saliency maps were compared to saliency masks manually extracted by a radiologist (A.A.). The agreement was evaluated as intersection area over the total area identified by the imager. DeGrave et al. [27] suggested that single local explanations are not enough to validate the correctness of a model against shortcuts and spurious correlations. Therefore, we propose a population-level explanation averaging the saliency maps of 200 randomly sampled images, with the prediction with the highest probability selected.

## 3. Results

In this section, we first introduce the baseline results (of the networks originally trained on CheXpert) and the performance of the networks following stacking, embedding, and fine tuning. Then, we present an in-depth analysis of the classification by applying Grad-CAM and comparing the extracted saliency maps with those generated by radiologists.

### 3.1. Baseline with Pretrained CNN

Table 5 shows the performance on the test set achieved by transfer learning without retraining in terms of AUC for each HUM-CXR class and on average. 

The results are shown for each CNN, ensembling with averaging, and weighted entropy averaging. Generally, the networks pretrained on CheXpert showed promising performance on the new dataset (HUM-CXR). Failures occurred mainly for bone. Ensembling generally achieved better average results compared to single-model performance.

### 3.2. Stacking and Embeddings

Combining the predictions with a metaclassifier (stacking) significantly improved bone classification and the mean classification AUC compared to the baseline. Furthermore, the embeddings extracted from pretrained CNNs were used to train tree-based classifiers. Table 6 shows the performance achieved by stacking and embeddings with DT, RF, and XRT ensembled with simple average and entropy-weighted average.

The best model (RF + simple averaging) achieved a mean AUC of 0.856 with a maximum of 0.94 for pleura. The results show that stacking and embedding achieved better classification performance compared to the baseline. Complex machine learning models (XRT and RF) achieved better performance than simple decision tree classifiers.

### 3.3. Fine Tuning

The last set of experiments consisted of fine tuning the classification layers of the pretrained CNNs (Table 7). Single-model performance improved with respect to transfer learning without retraining, except for VGG16 and VGG19. Ensemble AUC increased for all strategies. Fine tuning combined with stacking achieved the best AUC for PNX (0.97), whereas on average, it was performant than the best embedding model. However, these results show that fine tuning alone is not enough to achieve competitive performance, and an additional metaclassifier is required to combine the results. All the described models are available at https://github.com/DanieleLoiacono/CXR-Embeddings.

### 3.4. Grad-CAM

We averaged the saliency maps of two batches of 200 randomly sampled images computed with Grad-CAM. The Grad-CAM heat map emphasizes the salient area within the image in shades of red and yellow, whereas the rest of the image is colored in blues and greens. Figure 3 shows that at a population level, the model was generally focused on the lung field and did not take into account shortcuts or spurious correlations that could be present in the borders. 

We visualized the areas of the CXRs that the model predicted to be most indicative of each prediction using gradient-weighted class activation mappings (GradCAMs) [26] and by creating a bounding box surrounding it. Randomly selected examples are shown in Figure 4, Figure 5, Figure 6 and Figure 7.

In Figure 8, we superimposed the bounding boxes for two classes to show how the model looks at different input areas depending on the specific class.

Furthermore, we compared Grad-CAM maps with saliency masks extracted by a radiologist in terms of common area over the full area identified by the expert. Our models achieved an overall average agreement of 75% (80% lung, 65% pleura, 84% cardiac, 75% PNX, and 67% device), showing how the models automatically learned meaningful features from the images similarly to an expert radiologist. Explainable AI (XAI) algorithms for visualization are successful approaches to identify potential spurious shortcuts that the network may have learned. Overall, our CNNs focused on meaningful areas of the image for the respective prediction. We found some inconsistencies in some examples of device predictions, especially with pacemakers. Figure 9 shows an example of a correct classification but based on an area that does not match well the hardware of the CIED. 

The saliency map highlights the intracardiac leads as the region responsible for device prediction.

## 4. Discussion

In this work, we first developed and trained CNN models to extract features; thereafter, we proposed the application of different transfer learning approaches to the feature extractor stage of pretrained CNNs to a test dataset, proving the efficiency of transfer learning for domain and task adaptation in medical imaging. Finally, we used Grad-CAM saliency maps to interpret, understand, and explain CNNs and to investigate the presence of potential Clever Hans effects, spurious shortcuts, and dataset biases. Our results support the use of transfer learning to overcome the need for large datasets toward promising AI-powered medical imaging to assist imagers in automating repetitive tasks and prioritizing unhealthy cases. CNNs were first introduced in handwritten zip code recognition in [28], dramatically increasing the performance of deep learning models, especially with N-dimensional matrix input (e.g., three channels images). Since then, CNNs have proven successful capabilities for image analysis, understanding, and classification. Convolutional layers are used in sequence to progressively reduce the input size and simultaneously perform feature extraction, starting from simple patterns in early convolutional layers (edges, curves, etc.) to semantically strong high-level features in deeper layers. The feature maps, i.e., the output at each convolutional step, can be represented as a continuous vector that contains a low-dimensional representation of the image, namely the image embedding. Image embeddings meaningfully represent the original input in a transformed space, reducing the dimensionality. Image embeddings can be used as input to train classifiers based on trees, kernels, Bayesian statistics, etc. Thereby, the advantage of using embeddings lies in benefiting the feature extraction capabilities of CNNs trained on a large dataset of images while designing a specific classifier for new data and, eventually, for a slightly different task. We trained our CNNs with a large publicly available dataset [14] to create an efficient feature extractor that could learn from a large corpus of images. Next, we proposed three transfer learning approaches to apply the feature extractor stage of pretrained CNNs to a new local, independent dataset—HUM-CXRs. Transfer learning has shown remarkable capabilities in computer vision, boosting performance for applications with small datasets. Transfer learning avoids overfitting, in addition to enabling generalization from one task to another [13], although the generalization capabilities decrease according to the dissimilarity between the base task and target task. Transfer learning has been successful in several fields, including image classification [21,29,30], natural language processing [31,32,33,34], cancer subtype discovery [35], and gaming [36]. We applied transfer learning to medical imaging understanding and classification, envisioning the possibility of developing a library of pretrained models for different medical imaging modalities and tasks. Our first TL approach consisted of stacking the predictions of the pretrained CNNs and training an additional metaclassifier to learn the correspondence between them and the HUM-CXR outcomes. The second approach involved two steps: first, the image embeddings of the last convolutional layer were extracted, and additional tree-based classifiers were trained to classify them into the output vector. Finally, we applied a more conventional fine tuning of the last classification layer of each CNN. In this way, the classification layer was customized to the label vector of the new dataset, and the final weights were updated to learn the correspondence between the features extracted by the CNN and the output. In addition to achieving a best classification performance of 0.856 average AUROC, transfer learning with image embeddings has the advantage of minimizing the computational power, dataset dimensions, and time required to adapt the pretrained models to a new dataset and task. The time required to train our tree-based model was in the order of a few minutes, overcoming the need for considerable computational resources, long training times, and GPU availability.

As a proof of concept, we applied this framework to CXRs. CXRs are commonly used for diagnosis, screening, and prognosis; thus, large labeled datasets are already available, such as CheXpert [14], MIMIC-CXR [37], and ChestX-ray [38]. Several previous studies were focused on CXR diagnosis with deep learning, along with these publicly available datasets. CheXNet [39] achieved state-of-the-art performance on fourteen disease classification tasks with ChestXray data [38], and the modified version CheXNeXt [40] achieved radiologist-like performance on ten diseases. On the same dataset, Ye et al. [41] proposed localization of thoracic diseases, in addition to CXR classification. Along with the publication of the dataset, Irvin et al. [37] proposed a solution to achieve performance comparable to that of expert radiologists for the classification of five thoracic diseases. Recently, Pham et al. [23] improved state-of-the-art results on CheXpert, proposing an ensemble of CNN architectures. We used the same dataset as Irvin et al. [37], Pham et al. [23], and Giacomello et al. [15] for pretraining; however, whereas they focused on only five representative findings, we enlarged the classification to seven classes. We can compare the performance of cardiomegaly and pleural effusion, the two findings that are most similar between HUM-CXRs and CheXpert. With respect to cardiomegaly [14,15,23], achieved a best AUROC of 0.828, 0.854, and 0.910, respectively. With respect to pleural effusion, [14,15,23] achieved a best AUROC of 0.940, 0.964, and, 0.936, respectively. Our models obtained by transferring the knowledge acquired on CheXPert to an independent local dataset achieved a best AUROC of 0.88 and 0.97 for cardiac and pleura, respectively. However, [14,15,23] trained and tested on data from the same dataset, i.e., the same distribution, demographic and geographic characteristics (USA residents, Stanford Hospital) and—potential—bias in the data. Hence, these models are potentially prone to the ”Clever Hans” effect [42], which limits their actual transition to clinical application. Weber et al. [43] discussed the importance of evaluating the performance of a DL model for applications for which it was not explicitly trained to characterize its generalization capabilities and avoid the Clever Hans effect. Similarly, in a recent analysis of COVID-19 machine learning predictors, Roberts et al. [44] claimed that none of the works under review was reliable enough for the transition from scientific research to clinical routine due to dataset biases, insufficient model evaluation, limited generalizability, and lack of reproducibility, among other reasons. Furthermore, they argued that the scientific community is focusing too much on outperforming benchmarks on public datasets. Using only public datasets without generalizing to new data can lead to overfitting, strongly hindering clinical translation. For these reasons, in this work, we did not focus on outperforming the state of the art in CXR classification, instead proposing a reproducible framework to overcome some of the main limitations of DL in medical imaging toward a more robust AI-powered clinical routine. In particular, we achieved the following insights. The original models performed poorly on the baseline task (best average AUROC: 0.777), i.e., using the CNNs directly in inference on the new external independent dataset; therefore, even if they were trained on an extremely large dataset, the CNNs were not able to generalize to a new domain and additional data. On the other hand, using transfer learning, in particular with image embeddings, it is possible to adapt the original models to a new domain, i.e., a new hospital, geographic and demographic characteristics, and new tasks, i.e., different labels, with minimum effort and competitive performance (best average AUROC: 0.856). Our approach is not limited to our dataset and the highlighted application; it could be adopted and successfully applied by any other research group or hospital that might need to classify medical images but does not have either a sufficient volume of data or the computational resources to train the model. Following this framework, the resulting models will have excellent feature extraction capability learned from large public datasets, but they will be validated, tailored, and improved with respect to the specific application to achieve optimal results.

Although adherence to the FAIR principles [45] is recommended for scientific data management, a recent systematic review proved the scarce reproducibility of deep learning research. The majority of published deep learning studies focused on medical imaging were non-randomized retrospective trials (only 7% of prospective were tested in a real-world clinical setting) affected by a high risk of bias (72%), with a low adherence to existing reporting standards and without access to data and code (available in 5% and 7% of cases, respectively). Furthermore, deep learning studies typically scantly and elusively describe the used methods, affecting external validity and implementation in clinical settings [7]. To comply with the FAIR principles, respect legal requirements, and preserve the institutional policy, we exhaustively described our methods, providing details for each step, from image analysis to model building, and we made our models available (https://github.com/DanieleLoiacono/CXR-Embeddings) Regardless of the singular value of AUC for each class and the direct comparison between HUM-CXRs and CheXpert among labels, we demonstrated the efficiency of the proposed method. We believe that by making the data available, we guarantee the reproducibility of the proposed methodology, strongly encouraging other groups to repeat our approach with CXRs and/or other images (e.g., computed tomography (CT)).

Finally, CNNs are black-box models that are difficult to interpret, significantly hindering their acceptance in critical fields, such as medicine. Degrave et al. [27] demonstrated that DL models for COVID-19 detection relied on spurious shortcuts, such as lateral markers, image annotations, and borders, to distinguish between positive and negative patients instead of identifying real markers of COVID-19 in the lung field. They suggested that explainable AI (XAI) models should be applied to every AI application in medicine and should be a prerequisite for clinical translation to routine practice. The trustworthiness of AI models for clinical diagnosis and prognosis has to be accurately assessed before they can be applied in a real setting. Several algorithms have been proposed to overcome the intrinsic black-box nature of CNNs. DeepLIFT [46] and SHAP GradientExplainer [47] are based on feature importance, with the aim of measuring the relevance and importance of each input feature in the final predicted output, usually using the coefficients of linear models as interpretability models. Another proposed approach is the use of DGN-AM [48], which evaluates which neurons are maximally activated with respect to a particular input observation, with the aim of identifying input patterns that maximize the output activation. CAM [49], Grad-CAM [26], and LRP [50] create coarse localization maps of the important regions of the input defining the discriminative regions for a specific prediction.

For these reasons, we applied Grad-CAM [26] to our problem, with the aim of (1) interpreting, understanding, and explaining CNN black-box behavior through comprehensible explanations to increase the trust and acceptance in AI for medical imaging for translation to clinical routine; and (2) investigating the presence of potential Clever Hans effects, spurious shortcuts, and dataset biases. Overall, the explanations provided by Grad-CAM showed a satisfactory ability of the model to identify specific markers and features with respect to the identified class. Grad-CAM saliency maps were found inside the lung field, with particular attention to the correct side of the chest. Double-class images correctly showed the differences between chest findings. However, De Grave et al. [27] were skeptical about presenting only a few examples of explanations, as they may not truthfully represent the real behavior of the model. They discussed the need for a population-level explanation to demonstrate the correctness and reasoning of the entire model, in addition to selecting single examples. In this work, we presented randomly selected examples and population-level explanations averaging two batches of 200 CXRs. The averaged saliency maps presented in Figure 3 demonstrate a high level of attention in the center of the image, whereas the borders are almost useless. Our findings demonstrate that the models were generally focused on the lung field without deploying shortcuts and spurious correlations that may be present outside the lung field, such as annotations, different border dimensions, and lateral markers. Overall, examples of local explanations did not indicate the use of shortcuts as the general model. The only exception we identified concerns the device class, particularly when detecting a CIED. Whereas the model generally correctly focused on the hardware components, in some examples (Figure 9), it correctly classified ”device” but exploited the intracardiac leads. This finding is not incorrect, but we would expect the model to focus more on the hardware, i.e., the main box. We believe this might be caused by the original dataset on which the models were pretrained. The device class is extensive and includes lines, tubes, valves, catheters, CIEDs, hardware, coils, etc. However, the percentage proportion of each subclass is not publicly available, so it is possible that “some objects”, such as tubes, leads, electrodes, and catheters, are more present than CIEDs, inducing the model to focus on them. Furthermore, we investigated false-positive predictions with respect to the device class. In most cases, we assert that the model was correctly classified devices, although the ground truth was incorrect. The main reason for such false positives is that our labels were extracted from unstructured medical reports. Whereas diseases are clearly written and discussed in the report, cardiac devices, electrodes, prosthesis, and other “objects” may be omitted in the report because are not considered ”abnormal” as medical pathologies or clinically relevant. We reasonably believe that with further effort in the definition of the ground truth, the performance of “normal” and “device” labels can be improved.

In contrast to reports by Saporta et al. [51] and Arun et al. [52], who recently demonstrated the unreliability of current saliency methods to explain deep learning algorithms in chest X-ray interpretation, we proved a satisfactory match between Grad-CAM saliency maps and a human benchmark (overall average agreement of 75%), although our data confirmed the same issues (a larger gap between Grad-CAM and radiologist saliency maps in cases of diseases characterized by multiple instances, complex shapes, and small size [51]). We found more variability in some classes, such as pleura and device (65% and 67%, respectively), whereas lung, cardiac, and PNX exhibited greater confidence (80%, 84%, and 75%, respectively).

Results of our study may be of value for both the medical and the scientific communities, as well as for the general population. Overall, our results may impact AI applicability in the medical field, speeding up the grounding system of machine and deep learning algorithms toward clinical application, partially overcoming the problem of the increasing demand for medical doctors. In this work, we analyzed the adaptability and applicability of state-of-the-art imaging classification techniques to a new dataset of images collected in a different country with different scanners. Our models achieved competitive performances (AUC > 85%), correctly identifying and labeling seven classes from X-ray images. We also showed that our model correctly interpreted X-rays similarly to expert radiologists. We proved the feasibility of our approach to train large models and apply them in different countries and hospitals. Next steps of this work will include the investigation of more recent CNN models [53,54,55] and the validation of this proof of concept on other datasets, as well as on different kinds of images (e.g., computed tomography). Finally, we have to acknowledge some limitations in our study. First, our results are limited by the retrospective design of our study. Secondly, we did not evaluate the optimal transfer learning approach when we trained the seven CNNs on the CheXpert dataset; however, this was outside of the scope of the present work. Thirdly, different and more recent CNN models should also be used in future research.

## 5. Conclusions

In this work, we proposed three transfer learning approaches for medical imaging classification. We demonstrated that CNNs pretrained on a large public dataset of medical images can be exploited as feature extractors for a different task (i.e., different classes) and domain (different country, scanner, and hospital) than the original one. In particular, the extracted image embeddings contain relevant information to train an additional classifier with satisfactory performance on an independent local dataset. This overcomes the need for large private datasets, considerable computational resources, and long training times, which are major limitations for the successful applications of AI in clinical practice. Finally, we proved that we can rely on saliency map for deep learning explainability in medical imaging, showing that the models automatically learned how to interpret X-rays in agreement with expert radiologists.

## Figures and Tables

**Figure 1 diagnostics-12-02084-f001:**
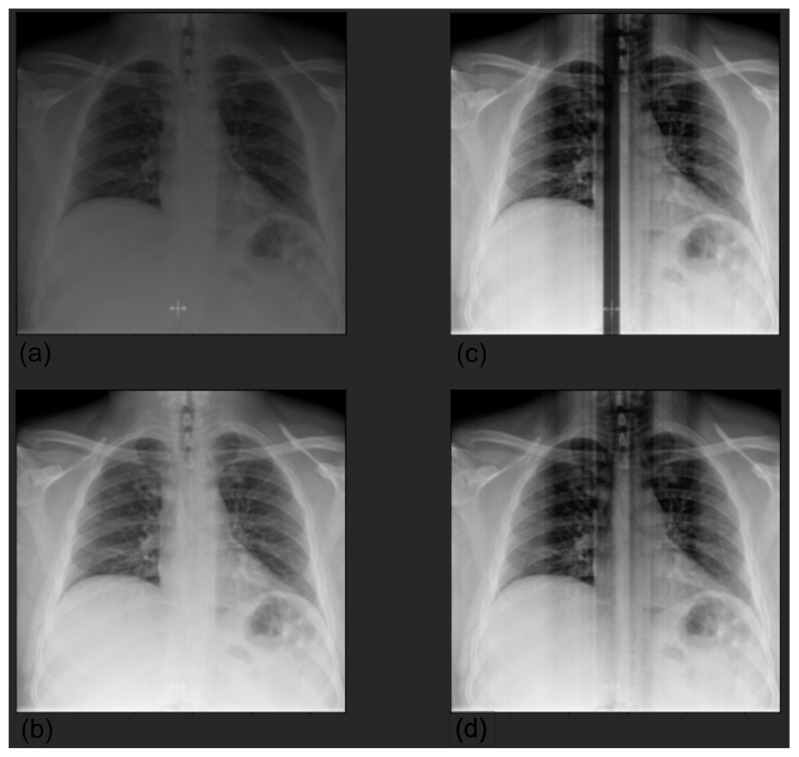
Preprocessing by clipping values larger than the 0.9995 quantile. The presence of a landmark, significantly whiter than the other pixels, created significant noise after normalization (**a**); original image (**b**); clipped image (**c**); normalized original image (**d**). To match the input dimension of the models, we resized the images to 224 × 224 and encoded them as RGB images by repeating the images for three channels. This was a necessary step in order to use the state-of-the-art image classification networks already pretrained on the ImageNet dataset. Then, we normalized each image by scaling the values in the range.

**Figure 2 diagnostics-12-02084-f002:**
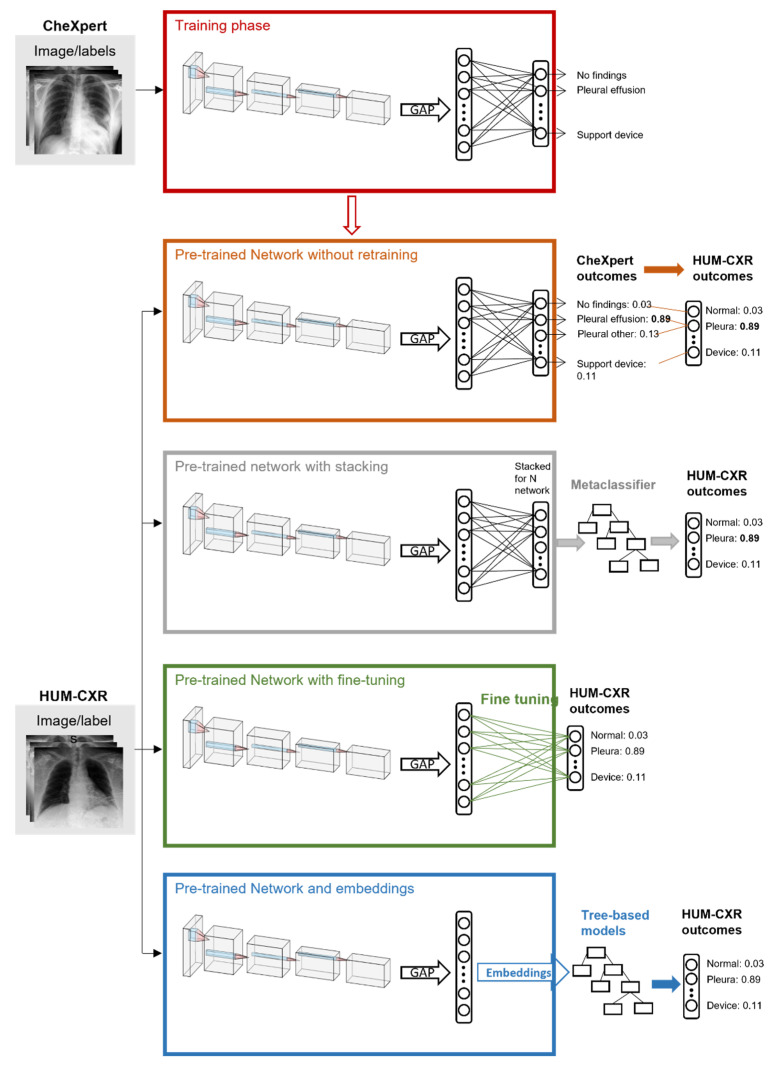
An overview of our experimental design. In the first phase, state-of-the-art image classification networks were tuned on a large public dataset of X-rays (CheXpert [14]). Then, we performed four different steps on the HUM-CXR dataset: (1) we tested the originally trained networks on the X-rays of the new dataset, mapping the HUM_CXR labels to CheXpert labels; (2) we used the originally pretrained networks with a metaclassifier to combine the predictions of each network on the new dataset; (3) we fine-tuned the networks by removing the fully connected classification layer from the seven CNNs trained on CheXpert and replacing it with a seven-output layer that matched HUM-CXR labels; and (4) we extracted the image embeddings from each network and trained tree-based classifiers to predict the HUM-CXR labels starting from the extracted embeddings.

**Figure 3 diagnostics-12-02084-f003:**
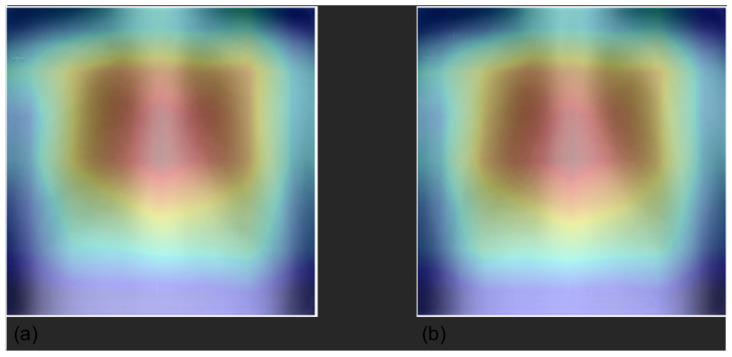
Average of Grad-CAM saliency maps for two batches (panels **a**,**b**) of 200 randomly sampled images. Images confirm that the model focuses on the lung field (image in shades of red and yellow) and does not take into account shortcuts or spurious correlations that could be present in the borders.

**Figure 4 diagnostics-12-02084-f004:**
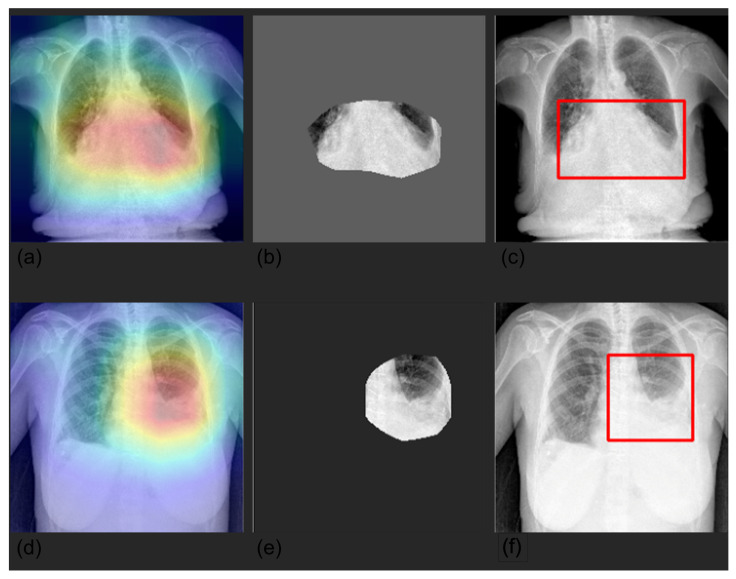
Visualization of pleura prediction maps for two selected CXRs. The panels represent the saliency mask obtained with Grad-CAM (panels **a**,**d**), the relevant area (mask values higher than the 0.8 quantile (panels **b**,**e**)), and the respective bounding box (panels **c**,**f**). The saliency mask focuses on plaura abnormalities, as shown by the heat map (panel **a**,**d**).

**Figure 5 diagnostics-12-02084-f005:**
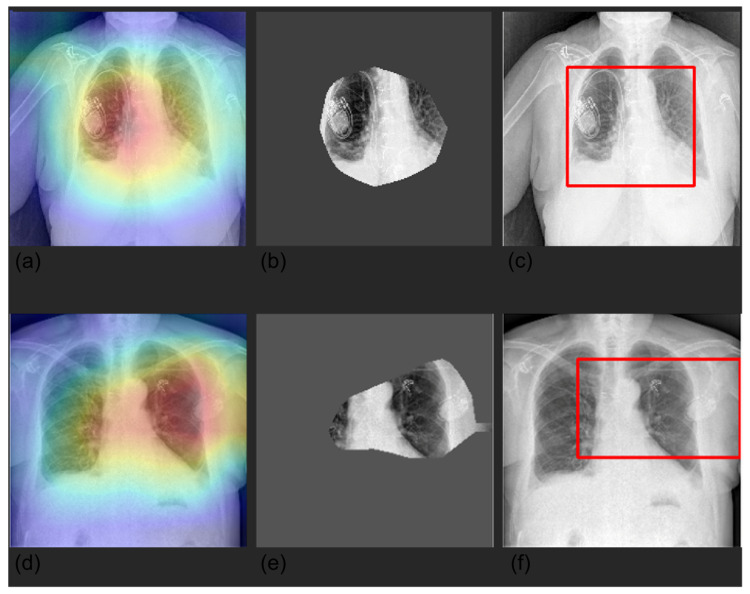
Visualization of device prediction maps for two selected CXRs. The panels represent the saliency mask obtained with Grad-CAM (panels **a****,****d**), the relevant area (mask values higher than the 0.8 quantile) (panels **b**,**e**), and the respective bounding box (panels **c**,**f**). The saliency mask focuses on device (hardware and/or leads), as shown by the heat map (panel **a**,**d**).

**Figure 6 diagnostics-12-02084-f006:**
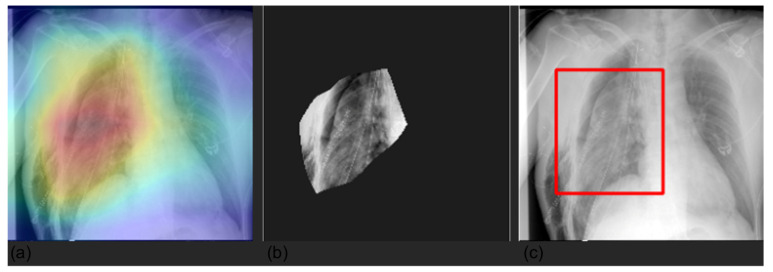
Visualization of pneumothorax prediction maps for a selected CXR. The panels represent the saliency mask obtained with Grad-CAM (panel **a**), the relevant area (mask values higher than the 0.8 quantile) (panel **b**), and the respective bounding box (panel **c**). The saliency mask, as emphasized by the heat map (panel **a**), focuses on the right lung field, which shows the pneumothorax.

**Figure 7 diagnostics-12-02084-f007:**
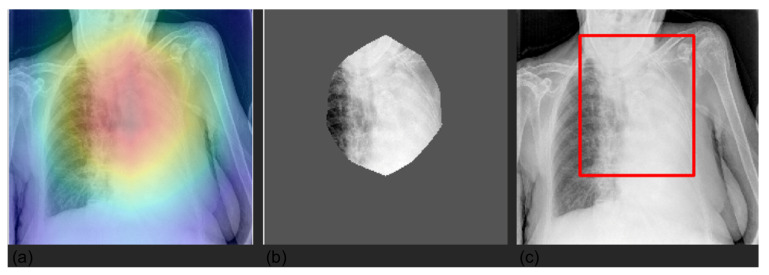
Visualization of lung prediction maps for a selected CXR. The panels represent the saliency mask obtained with Grad-CAM (panel **a**), the relevant area (mask values higher than the 0.8 quantile) (panel **b**), and the respective bounding box (panel **c**). The saliency mask, as emphasized by the heat map (panel **a**), focuses on the left lung, which shows lung abnormality.

**Figure 8 diagnostics-12-02084-f008:**
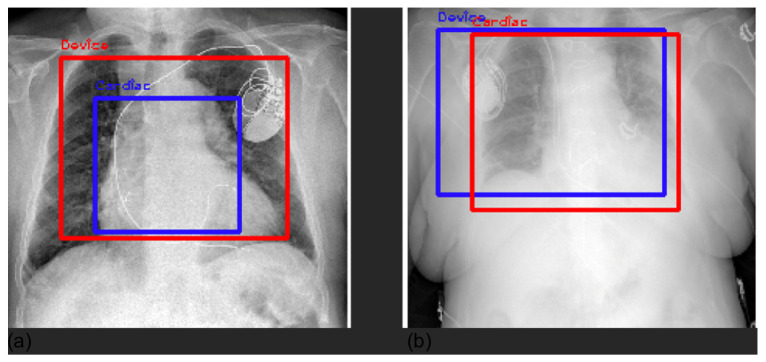
Superimposition of bounding boxes for cardiac (panel **a**, cardiac in blue and device in red) and device (panel **b**, device in blue and cardiac in red) outcomes for two examples.

**Figure 9 diagnostics-12-02084-f009:**
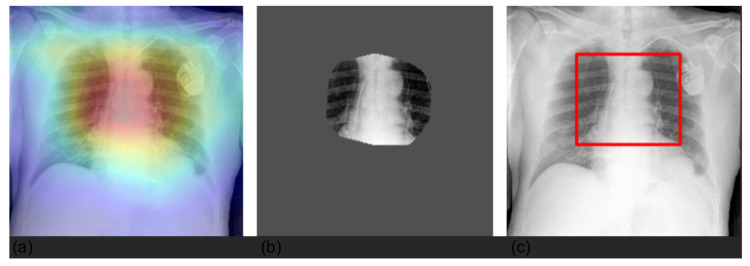
Shortcut for the identification of a pacemaker focusing on the leads: the saliency mask obtained with Grad-CAM (panel **a**), the relevant area (mask values higher than the 0.8 quantile) (panel **b**), and the respective bounding box (panel **c**).

**Table 1 diagnostics-12-02084-t001:** Absolute frequencies of positive, uncertain, and negative samples for each finding (relative frequencies are reported in parentheses) in the CheXpert dataset (*n* = 191,027).

Label	Positive (%)	Uncertain (%)	Negative (%)
No Finding	16,974 (8.89)	0 (0.0)	174,053 (91.11)
Enlarged card.	30,990 (16.22)	10,017 (5.24)	150,020 (78.53)
Cardiomegaly	23,385 (12.24)	549 (0.29)	167,093 (87.47)
Lung opacity	137,558 (72.01)	2522 (1.32)	50,947 (26.67)
Lung lesion	7040 (3.69)	841 (0.44)	183,146 (95.87)
Edema	49,675 (26.0)	9450 (4.95)	131,902 (69.05)
Consolidation	16,870 (8.83)	19,584 (10.25)	154,573 (80.92)
Pneumonia	4675 (2.45)	2984 (1.56)	183,368 (95.99)
Atelectasis	29,720 (15.56)	25,967 (13.59)	135,340 (70.85)
Pneumothorax	17,693 (9.26)	2708 (1.42)	170,626 (89.32)
Pleural effusion	76,899 (40.26)	9578 (5.01)	104,550 (54.73)
Pleural other	2505 (1.31)	1812 (0.95)	186,710 (97.74)
Fracture	7436 (3.89)	499 (0.26)	183,092 (95.85)
Support devices	107,170 (56.1)	915 (0.48)	82,942 (43.42)

**Table 2 diagnostics-12-02084-t002:** Absolute frequencies of positive, uncertain, and negative samples for each finding (relative frequencies are reported in parentheses) in the HUM-CXR dataset (*n* = 941).

Label	Positive (%)	Negative (%)
Normal	273 (29.01)	668 (70.99)
Cardiac	93 (9.88)	848 (90.12)
Lung	427 (45.38)	514 (54.62)
Pneumothorax	38 (4.04)	903 (95.96)
Pleura	135 (14.35)	806 (85.65)
Bone	137 (14.6)	804 (85.4)
Device	147 (15.56)	794 (84.44)

**Table 3 diagnostics-12-02084-t003:** Correspondence between CheXpert and HUM-CXR labels.

CheXpert	HUM-CXR
Pleural effusion, pleural other	Pleura
Support devices	Device
Pneumothorax	PNX
Enlarged cardiomediastinum, cardiomegaly	Cardiac
Lung opacity, lung lesion, consolidation, pneumonia, atelectasis, edema	Lung
Fracture	Bone
No findings	Normal

**Table 4 diagnostics-12-02084-t004:** Embedding model hyperparameters.

Model	Hyperparameters
DT	Max depth = [1, 2, 3, 4, 5, 10, 20], min samples leaf = [1, 2, 4], min samples split = [2, 5, 10], criterion = [gini, entropy] **Final values:** Max depth = 10, min samples leaf = 1, min samples split = 2, criterion = gini
RF	Max depth = [1, 2, 3, 4, 5, 10, 20], min samples leaf = [1, 2, 4], min samples split = [2, 5, 10], criterion = [gini, entropy], number estimators = [10, 20, 30, 50, 100, 200, 300] **Final values:** Max depth = 10, min samples leaf = 4, min samples split = 10, criterion = gini, number of estimators = 100
XRT	Max depth = [1, 2, 3, 4, 5, 10, 20], min samples leaf = [1, 2, 4], min samples split = [2, 5, 10], criterion = [gini, entropy], number estimators = [10,20,30, 50, 100, 200, 300] **Final values:** Max depth = 10, min samples leaf = 2, min samples split = 2, criterion = entropy, number of estimators = 200

**Table 5 diagnostics-12-02084-t005:** CNN results with pretrained networks without retraining in terms of AUC. Each column represents an HUM-CXR label. We report the results for each network and for the two ensembling strategies. The best results for each class and average are highlighted in bold.

Model	Normal	Cardiac	Lung	PNX	Pleura	Bone	Device	Mean
DenseNet121	0.81	0.84	0.70	0.89	0.87	0.39	0.87	0.766
DenseNet169	0.80	0.79	0.69	0.90	0.87	0.36	0.88	0.755
DenseNet201	0.81	0.78	0.70	0.90	0.87	0.35	0.86	0.754
InceptionResNetV2	0.81	0.83	0.69	0.89	0.87	0.39	0.86	0.762
Xception	0.80	0.77	0.69	**0.91**	0.87	0.44	0.86	0.764
VGG16	0.82	**0.85**	0.70	0.89	0.89	0.41	0.86	0.775
VGG19	0.81	0.83	0.71	0.88	0.89	0.42	0.85	0.772
Averaging	0.82	0.84	0.71	**0.91**	0.89	0.38	**0.89**	0.777
Entropy	0.82	0.83	0.71	**0.91**	0.89	0.37	**0.89**	0.772

**Table 6 diagnostics-12-02084-t006:** Results of stacking and tree-based models trained on embeddings extracted from pretrained CNNs in terms of AUC. Each column represents an HUM-CXR finding. We report the results for each tree model and for both ensembling strategies. Best results for each class and average are highlighted in bold.

Model	Normal	Cardiac	Lung	PNX	Pleura	Bone	Device	Mean
Stacking	0.85	0.81	**0.74**	0.88	**0.94**	**0.85**	0.84	0.843
DT+averaging	0.81	0.69	0.68	0.75	0.88	0.68	0.78	0.734
RF+averaging	**0.86**	**0.85**	0.72	**0.92**	**0.94**	**0.85**	**0.86**	**0.856**
XRT+averaging	0.85	0.84	0.73	**0.92**	**0.94**	**0.85**	0.85	0.853
DT+entropy	0.81	0.69	0.68	0.75	0.88	0.69	0.78	0.753
RF+entropy	0.85	**0.85**	0.72	**0.92**	**0.94**	**0.85**	0.85	0.853
XRT+entropy	0.85	0.84	0.73	**0.92**	**0.94**	**0.85**	0.84	0.852

**Table 7 diagnostics-12-02084-t007:** CNN results with fine tuning of the classification layer of pretrained networks in terms of AUC. Each column represents an HUM-CXR finding. We report the results for each single network, for the two ensembling strategies, and for stacking. The best results for each class and average are highlighted in bold.

Model	Normal	Cardiac	Lung	PNX	Pleura	Bone	Device	Mean
DenseNet121	**0.81**	0.73	0.76	0.94	0.90	0.73	0.78	0.807
DenseNet169	0.73	**0.88**	0.77	0.95	0.94	0.72	0.72	0.814
DenseNet201	0.83	0.81	0.71	0.94	0.94	0.74	0.82	0.828
InceptionResNetV2	**0.81**	0.86	**0.79**	0.90	0.93	0.69	0.76	0.818
Xception	**0.81**	0.82	0.73	0.94	0.95	0.68	0.80	0.819
VGG16	0.67	0.81	0.72	0.33	0.95	0.62	0.82	0.704
VGG19	0.64	0.79	0.44	0.83	0.93	0.52	**0.87**	0.717
Averaging	0.83	0.86	0.78	**0.96**	0.96	0.71	0.81	0.842
Entropy	**0.81**	0.86	0.78	0.95	0.96	0.73	0.83	0.845
Stacking	0.80	0.85	0.74	0.93	**0.97**	**0.83**	0.86	**0.853**

## Data Availability

This manuscript represents valid work, and neither this manuscript nor one with substantially similar content under the same authorship has been published or is being considered for publication elsewhere. Arturo Chiti had full access to all the data in the study and takes responsibility for the integrity of the data and the accuracy of the data analysis. All the described models are available at https://github.com/DanieleLoiacono/CXR-Embeddings.

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
