# Peer review of "Image Embeddings Extracted from CNNs Outperform Other Transfer Learning Approaches in Classification of Chest Radiographs"

_diagnostics, 2022, doi:10.3390/diagnostics12092084_

Round 1

Reviewer 1 Report

The paper is generally fine; however, I have the following comments.

How did you convert the X-ray images from one channel to three channels?

There are some recent CNN models. You can also investigate them. There is a very recent relevant paper:  “Tuberculosis Detection in Chest Radiograph Using Convolutional Neural Network Architecture and Explainable Artificial Intelligence,” Neural Computing and Applications, 2022. DOI: 10.1007/s00521-022-07258-6

How about recent transformers? (1) Xie, E., Wang, W., Yu, Z., Anandkumar, A., Alvarez, J. M., & Luo, P. (2021). SegFormer: Simple and efficient design for semantic segmentation with transformers. Advances in Neural Information Processing Systems, 34. (2) Liu, Z., Lin, Y., Cao, Y., Hu, H., Wei, Y., Zhang, Z., ... & Guo, B. (2021). Swin transformer: Hierarchical vision transformer using shifted windows. In Proceedings of the IEEE/CVF International Conference on Computer Vision (pp. 10012-10022). 

'The data were divided 209 into 70% training set and 30% test set.' Why did you not use the cross-validation approach?

The hyperparameter values are not defined.

A confusion matrix of the system should be provided.

The Kappa value or the F1 score should be provided. 

Author Response

Comments and Suggestions for Authors

The paper is generally fine; however, I have the following comments.

How did you convert the X-ray images from one channel to three channels?

We clarified this step in the paper: we encoded them as RGB images, repeating the one-channel image for three channels. Despite such an encoding will lead to some redundancy, it was necessary (as mentioned in the paper) to match the input shape of the models pre-trained on ImageNet that have been used in our work. In this was it is possible to exploit the pre-training step of state-of-the-art image classification network and fine tune them for the specific applications.

There are some recent CNN models. You can also investigate them. There is a very recent relevant paper:  “Tuberculosis Detection in Chest Radiograph Using Convolutional Neural Network Architecture and Explainable Artificial Intelligence,” Neural Computing and Applications, 2022. DOI: 10.1007/s00521-022-07258-6

Please see the answer below.

How about recent transformers? (1) Xie, E., Wang, W., Yu, Z., Anandkumar, A., Alvarez, J. M., & Luo, P. (2021). SegFormer: Simple and efficient design for semantic segmentation with transformers. Advances in Neural Information Processing Systems, 34. (2) Liu, Z., Lin, Y., Cao, Y., Hu, H., Wei, Y., Zhang, Z., ... & Guo, B. (2021). Swin transformer: Hierarchical vision transformer using shifted windows. In Proceedings of the IEEE/CVF International Conference on Computer Vision (pp. 10012-10022). 

We thank the reviewer for these references. We do agree that applying transformers to computer vision tasks is a very promising research direction. However, we would like to stress that the major goal of our paper is to investigate how to exploit and adapt models pre-trained on large public datasets to solve a similar task on a smaller private dataset. Accordingly, as explained in the paper, we used “the most common architecture used to perform classification”. Undoubtfully, there are also other CNN networks that can be successfully applied to CXR images (as the EfficientNet that in the experiment of Nafisah et al. outperformed some of the networks used in our work). Nevertheless, as we stated in the paper “each architecture presented different performances on different labels”, suggesting that none of the network could easily outperform all the others on all the possible datasets/tasks (as an example, the results reported in Nafisah et al. are specific for tuberculosis, an infection not frequent in the Italian general population). Therefore, we believe that commonly used CNN models are the best choice for the goal of our work (i.e., investigating how adapting pre-trained models to new tasks). Especially, this increases the robustness of the classification for all labels towards clinical translation and use, showing the applicability to new datasets and hospitals. We included additional comments about using more recent CNN/transformers models in the discussion section.

'The data were divided 209 into 70% training set and 30% test set.' Why did you not use the cross-validation approach?

We thank the reviewer for this question which give us the possibility to comment this aspect. Whilst cross-validation is a typical approach to test the performance of machine learning models, especially for hyperparameter tuning and/or small dataset, it is also quite common, in case of large dataset to divide the dataset in training and test. Indeed, we used cross-validation to tune the hyperparameters of the ML models, as we also better clarified this aspect in the revised version of the main text. We believe that, regarding the final performances, in our case a training-test validation is already robust enough to test the models on the new dataset. Furthermore, since our goal was not to introduce new innovative x-ray classification networks but rather show the adaptability and applicability of common networks for new datasets for future clinical translation, the cross-validation step is a computational consuming not necessary step, especially for fine-tuning the CNN networks, since after finding the best model they still need to be re-tuned on a training set and used in inference for new data. Therefore, for fair and homogeneous comparison between stacking, single models, fine tuning and embeddings, we decided to perform a training-test validation.

The hyperparameter values are not defined.

In the original draft we reported only the final values of the hyperparameters, In the revised version we added all the tested values (in Tab 4).

A confusion matrix of the system should be provided.

In the task considered in this paper, any sample could be annotated with multiple labels (i.e., each sample might belong to more than one class). This makes confusion matrix quite difficult to read unless computed independently for each label like several binary classification problems. For this reason, we decided to stick with the AUC that is the standard metrics used in the literature for evaluating the performance on the CheXpert dataset, that is similar to the HUM-CXR dataset used in this work.

The Kappa value or the F1 score should be provided. 

These scores are computed for a specific threshold of the classifier, while the benefit of AUC is to provide an overview of the classifier performance for different possible thresholds.

Reviewer 2 Report

-The paper should be interesting ;;;

-it is a good idea to add a block diagram of the proposed research (step by step);;;

-it is a good idea to add more photos of measurements, sensors + arrows/labels what is what (If any);;;

-What is the result of the analysis?;;

-figures should have high quality;;; 

-labels of figures should be bigger ;;;

-please add photos of the application of the proposed research, 2-3 photos ;;; 

-what will society have from the paper?;;

-please compare advantages/disadvantages other approaches;;;

-Conclusion: point out what have you done;;;;

-please add some sentences about future work;;;

Author Response

Comments and Suggestions for Authors

-The paper should be interesting ;;;

-it is a good idea to add a block diagram of the proposed research (step by step);;;

Figure 2 summarizes our research plan. We added an in-depth description of the proposed research in the caption of the Figure to better explain each step.

-it is a good idea to add more photos of measurements, sensors + arrows/labels what is what (If any);;;

We are not sure what the reviewer is referring to, as our work does not involve neither measurements nor sensors.

-What is the result of the analysis?;;

We modified sections 3, 4, and 5, giving more emphases to our results.

-figures should have high quality;;; -labels of figures should be bigger ;;;

High-resolution figures, modified as requested, will be provided if the paper will be accepted.

-please add photos of the application of the proposed research, 2-3 photos ;;; 

We included in our paper both figures to describe visually our models/approaches and figures to show the results achieved with our classifieds. There are no relevant “photos” we could add to the paper.

-what will society have from the paper?;

We thank the reviewer for this question. As presented in the introduction, today’s society is affected by an exponential population growth linked with aging of the population. Aging population requires larger attention from the healthcare system, including high demands for radiologists and imagers. However, there is currently a gap in radiologists and AI might be partially fill this gap. In this work, we analyzed the adaptability and applicability of state-of-the-art imaging classification techniques to a new dataset, in a different country, collected with different scanners. Our models achieved competitive performances  (more than 85% AUC) in detecting seven different classes from x-rays. We also showed that our model correctly interpreted x-rays similarly to expert radiologists. Our approach shows the feasibility to train large models and apply them in different countries and hospitals, being an added value for doctors and filling the gap between the need of imaging and the available doctors. We modified the paper commenting this aspect in the revised version of the discussion section. 

-please compare advantages/disadvantages other approaches;;;

In the state-of-the-art there are several approaches to analyze medical images. CNNs are the most common approach for detection, classification and segmentation (e.g., Ronnenberg et al, 2015). More recent options include the use of transformers to computer vision tasks. However, as explained in the paper, we used “the most common architecture used to perform classification”, to exploit the power of having large pre-trained networks easily adaptable and applicable. Undoubtfully, there are also other CNN networks that can be successfully applied to CXR images (as the EfficientNet that in the experiment of Nafisah et al. outperformed some of the networks used in our work). Therefore, we believe that state-of-the-art CNN image classification networks are the best choice for the goal of our work (i.e., investigating how adapting pre-trained models to new tasks). As suggested by the review, we included further comments in the discussion section.

-Conclusion: point out what have you done;;;;

We thank the reviewer for this comment, and we modified accordingly the conclusion section of the revised version of the manuscript.

-please add some sentences about future work;;;

We modified accordingly the dicussion section of the revised version of the manuscript.

Round 2

Reviewer 1 Report

Thank you for addressing my comments.

Author Response

Thank you for your comment.

Reviewer 2 Report

-arrows to figures 5-9 should be added (what is what). The red color and yellow color should be explained.

-block diagram of research + arrows should be added;; it is missing?

Author Response

We thank the reviewer for comments on the images. Anyway, in our opinion arrows wouldn't clarified figures 5-9 since the salient part of images is not represented by a particular point, but an area. This was the reason because we have build each figure with three panels: a) the saliency mask obtained with Grad-CAM (panel a), b) the relevant area (mask values larger than 0.8 quantile) (panel b), and c) the respective bounding box (panel c). We added the meaning of colours within the saliency mask identified by the Grad-CAM (lines 337-338). The heatmap emphasizes the salient area within the image in shades of red and yellow, while the rest of the image is colored in blues and greens. 

Block diagram and arrows in figure 2 are present, they will be more clear in the high-quality version of the image.